# Synthesis, Antimicrobial, Anti-Virulence and Anticancer Evaluation of New 5(4*H*)-Oxazolone-Based Sulfonamides

**DOI:** 10.3390/molecules27030671

**Published:** 2022-01-20

**Authors:** Ahmad J. Almalki, Tarek S. Ibrahim, Ehab S. Taher, Mamdouh F. A. Mohamed, Mahmoud Youns, Wael A. H. Hegazy, Amany M. M. Al-Mahmoudy

**Affiliations:** 1Department of Pharmaceutical Chemistry, Faculty of Pharmacy, King Abdulaziz University, Jeddah 21589, Saudi Arabia; tmabrahem@kau.edu.sa; 2Center of Excellence for Drug Research and Pharmaceutical Industries, King Abdulaziz University, Jeddah 21589, Saudi Arabia; 3Department of Pharmaceutical Organic Chemistry, Faculty of Pharmacy, Al-Azhar University, Assiut 71524, Egypt; ehabtaher@azhar.edu.eg; 4Department of Pharmaceutical Chemistry, Faculty of Pharmacy, Sohag University, Sohag 82524, Egypt; mamdouh.fawzi@pharm.sohag.edu.eg; 5Department of Biochemistry and Molecular Biology, Faculty of Pharmacy, Helwan University, Cairo 11795, Egypt; dr.mahmoudyouns@yahoo.com; 6Department of Microbiology and Immunology, Faculty of Pharmacy, Zagazig University, Zagazig 44519, Egypt; waelmhegazy@daad-alumni.de; 7Department of Pharmaceutical Organic Chemistry, Faculty of Pharmacy, Zagazig University, Zagazig 44519, Egypt

**Keywords:** oxazolone, sulfonamide, antimicrobial, anti-virulence, antibiofilm, anticancer

## Abstract

Since the synthesis of prontosil the first prodrug shares their chemical moiety, sulfonamides exhibit diverse modes of actions to serve as antimicrobials, diuretics, antidiabetics, and other clinical applications. This inspiring chemical nucleus has promoted several research groups to investigate the synthesis of new members exploring new clinical applications. In this study, a novel series of 5(4*H*)-oxazolone-based-sulfonamides (OBS) **9a**–**k** were synthesized, and their antibacterial and antifungal activities were evaluated against a wide range of Gram-positive and -negative bacteria and fungi. Most of the tested compounds exhibited promising antibacterial activity against both Gram-positive and -negative bacteria particularly OBS **9b** and **9f**. Meanwhile, compound **9h** showed the most potent antifungal activity. Moreover, the OBS **9a**, **9b**, and **9f** that inhibited the bacterial growth at the lowest concentrations were subjected to further evaluation for their anti-virulence activities against *Pseudomonas aeruginosa* and *Staphylococcus aureus*. Interestingly, the three tested compounds reduced the biofilm formation and diminished the production of virulence factors in both *P. aeruginosa* and *S. aureus*. Bacteria use a signaling system, quorum sensing (QS), to regulate their virulence. In this context, in silico study has been conducted to assess the ability of OBS to compete with the QS receptors. The tested OBS showed marked ability to bind and hinder QS receptors, indicating that anti-virulence activities of OBS could be due to blocking QS, the system that controls the bacterial virulence. Furthermore, anticancer activity has been further performed for such derivatives. The OBS compounds showed variable anti-tumor activities, specifically **9a**, **9b**, **9f** and **9k,** against different cancer lines. Conclusively, the OBS compounds can serve as antimicrobials, anti-virulence and anti-tumor agents.

## 1. Introduction

The increasing pervasiveness of microbial resistance represents a major issue globally. Despite the discovery and significant development of several new antibiotics, multidrug-resistant bacteria are still becoming more prevalent, and are creating a serious public health risk for the population [1,2]. It has been estimated that 700,000 people in the world die every year from antibiotic-resistant infectious bacterial diseases. In the absence of new prevention or treatment remedies, by 2050, it is estimated that 10 million people worldwide will die of these infectious diseases each year [3]. Consequently, the development of a new, powerful therapeutic approach to treat and kill Gram-negative, as well as Gram-positive, human pathogens is urgently needed. It is well-recognized that antibiotics-resistant bacterial infections are not due to free bacteria but rather to bacteria existing within a biofilm [4]. The resistance of biofilm-forming bacteria to conventional antimicrobials is attributed to: (1) the failure of the antimicrobial to penetrate the biofilm, (2) the evolution of complex drug resistance properties, and (3) biofilm mediated inactivation or modification of antimicrobial enzymes [5].

Conversely, heterocycles are a significant class of cyclic compounds that are considered the prominent source of biologically active compounds due to their diverse structures [6,7,8]. Among them, oxazol-5-(4*H*)-one (2,4-disubstituted 5-oxo-4,5-dihydro-1,3-oxazole, also known as azalactone or oxazolone) is one of the large varieties of interesting molecules with numerous applications in chemistry and biology [9,10]. The exocyclic double bond in position four of the oxazolone ring provides a new reactivity that allows the construction of interesting derivatives [10]. Moreover, 2-phenyloxazol-5(4*H*)-ones with an additional exocyclic double bond exhibits a wide range of biological activities such as antibacterial [11], immunomodulatory [10,12,13], antidiabetic [14], antiviral [15], antifungal [16], anticancer [17], anti-inflammatory [18], anti-HIV [19], anti-angiogenic [20], sedative [21], and tyrosinase inhibitory activities [22], among others (Figure 1). Notably, there are numerous drugs containing oxazolone motif in their structure such as the carbonate codrug, CB-NTXOL-BUPOH (**I**), consisting of 6-β-naltrexol (the major active metabolite of naltrexone, a potent μ-opioid receptor antagonist used in the treatment of alcohol dependence and opioid abuse) covalently linked by carbonate ester linkage to a modified form of hydroxybupropion (bupropion with oxazolone) [23]. Deflazacort (**II**) has anti-inflammatory and immunosuppressive effects [24]. ZHD-0501 (**III**) is a metabolite of staurosporine (STA) analog with an oxazolone scaffold, which inhibits the proliferation of several human and murine cancer cell lines [25]. Jadomycin (**IV**) is an antifungal with a unique 8*H*-benz[b]oxazolo [3,2-] phenanthridine pentacyclic skeleton produced by the bacterium *Streptomyces venezuelae* ISP 5230 [26]. Posizolid (**V**) is an oxazolidinone antibiotic under investigation by AstraZeneca for the treatment of bacterial infections [27]. Moreover, oxazolone scaffold is an attractive heterocyclic precursor which can be used as versatile building blocks in organic synthesis, as they consist of “masked” amino acids and contain numerous reactive sites allowing a diversity of possible modifications. Their reactivity (nucleophilic attack to the carbon atom at position five of the oxazolone ring) makes them excellent substrates for the preparation of structurally complex amino acids and highly substituted heterocycles, enol acetate and benzoxazinone derivatives, phenylpyruvic acid, imidazolinones, amino acids, amino alcohols, amides, dyes and triazinones [9,10,28]. The azlactone transformations have allowed facile access to natural compounds and pharmaceutically and biologically intriguing molecules.

Furthermore, sulfonamide derivatives have evoked high favor and constitute privileged structural motifs in medicinal chemistry because they exhibit a wide range of pharmacological activities [8] including anticancer [29,30], antibacterial [31], anti-tuberculosis [32], anti-carbonic anhydrase [33], anti-fungal [34], anti-inflammatory [35], anti-diabetic [36], antiviral [37], anti-oxidant [38], diuretic [36], antimalarial [39], and antithyroid [36], in addition to protease inhibitory activity in vitro and in vivo, among others [36]. Obviously, some sulfonamide derivatives have been approved by FDA for cancer therapy. For instance, the third approved histone deacetylase (HDAC) inhibitor, Belinostat (**VI**), is approved to treat T-cell lymphoma after Vorinostat and Romidepsin (Figure 2) [40]. The topoisomerase II inhibitor, Amsacrine (**VII**), is approved to treat acute leukemias and malignant lymphomas through intercalating into the DNA of tumor cells (Figure 2) [41]. Additionally, the highly selective Bcl-2 inhibitor, Venetoclax (**VIII**), is now approved for treatment of chronic lymphocytic leukemia (CLL) patients with a 17p chromosomal deletion who have received at least one prior therapy (Figure 2) [42,43]. Moreover, sulfonamide moiety is usually considered as an effective bioisostere of the carboxylic group because the distance between two oxygen atoms is about similar in these two functional groups [44,45]. Therefore, sulfonamide motif could be engaged in a network of hydrogen bonds which are the same as the carboxylic group with fewer drawbacks of the carboxylic group, such as metabolic instability, toxicity, and limited passive diffusion across biological membranes [44,45].

Motivated by the above information and based on bacterium being linked to cancer by two mechanistic pathways—induction of chronic inflammation and production of carcinogenic bacterial metabolites [46]—a series of new 5(4*H*)-oxazolone-benzene sulfonamide derivatives were designed, synthesized and evaluated for their antibacterial, antifungal, antibiofilm, anti-virulence and anticancer activities. Moreover, a molecular docking study was carried out to investigate the binding mode and interaction of the most potent derivatives into the active site of *Pseudomonas aeruginosa* quorum sensing (QS) receptor (PDB: 1ROS) that orchestrates the bacterial virulence [47,48].

## 2. Results and Discussion

### 2.1. Chemistry

The chemical synthetic approach of the target compounds **9a**–**k** is outlined in Figure 1 and Figure 2 As illustrated in Figure 1. 4-Toluenesulfonyl anthranilic **3** was synthesized according to the reported protocol via nucleophilic substitution reaction of anthranilic acid **1** with *p*-toluensulfonyl chloride **2** in the presence of sodium hydroxide [47]. Compound **3** was subjected to the same reaction with 1*H*-benzotriazole **4** in DCM using thionyl chloride at room temperature to afford the benzene sulfonamide **5** in excellent yield. The latter compound was further treated with glycine **6** in acetonitrile/H_2_O under basic conditions to furnish (2-((4-methylphenyl)sulfonamido)benzoyl) glycine **7** (Figure 1).

The former acid **7** was reacted with the appropriate aldehydes **8a**-**j** using acetic anhydride in the presence of anhydrous sodium acetate to give the desired oxazolone-benzenesulfonamide derivatives **9a**–**k** (Figure 2). All the synthesized compounds were in accordance with their expected structures which have been elucidated by various spectroscopic techniques such as ^1^H NMR and ^13^C NMR spectra and elemental analyses (see Appendix A).

### 2.2. Biological Screening

#### 2.2.1. In Vitro Antimicrobial Activity

##### Minimum Inhibitory Concentrations (MICs) of Synthesized Compounds against Different Gram-Positive and -Negative Bacteria

In order to evaluate the antimicrobial activities of synthesized compounds, their MICs were determined against different Gram-positive and -negative bacteria, and fungi (Table 1). Most of the tested compounds exhibited promising antibacterial activity. Compounds **9a**, **9b**, **9c**, **9e** and **9f** were the most active derivatives with broad spectrum of activity against Gram-positive and Gram-negative bacteria. Among them, **9a** (with unsubstituted phenyl group), **9b** (4-methoxy) and **9f** (4-NO_2_) were the most potent. The result of antifungal activity screening showed that most of the tested derivatives had moderate, or weak activity, or were inactive against the used fungal strains, as illustrated in Table 1. Ongoing throughout the details, **9a** (with unsubstituted phenyl group) showed a broad spectrum of activity against all bacterial strains, in particular against *Escherichia coli*. However, **9a** had weak antifungal activity against *Aspergillus niger* and moderate antifungal activity against *Candida albicans*.

Replacement of hydrogen with the electron donating methoxy group, i.e., OBS **9b,** resulted in a 2-fold increase in the activity against *Staphylococcus aureus*, *Staphylococcus epidermidis*, *Micrococcus* spp. and *Pseudomonas aeruginosae* and retained the same activity as OBS **9a** against *Klebsiella pneumonia*, *Salmonella typhimurium* and *Escherichia coli*. Introduction of 2,5-dimethoxy group, as in OBS **9c,** retained the same activity against *S. epidermidis*, and *Micrococcus* spp. with a 2-fold decrease against *S. aureus*, *P. aeruginosa*, *K. pneumonia*, *S. typhimurium* and *E. coli*. Replacement of hydrogen with trimethoxy groups, i.e., OBS **9d,** retained the same activity as **9a** against *S. aureus* and improved (2-fold) activity against *S. epidermidis* and *Micrococcu89s* spp. against compound **9a**. Moreover, compound **9d** exhibited a slight decrease in activity (2-fold) against *E. coli* and a marked decrease in the antibacterial activity against *P. aeruginosa*, *K. pneumonia*, and *S. typhimurium*. Introduction of the weakly deactivating Cl group, as in compound **9e**, retained the broad spectrum of activity against all Gram-positive and -negative bacteria with improved potency (2-fold increase in activity) against all Gram-positive organisms: *S. epidermidis*, and *Micrococcus* spp. Additionally, compound **9e** displayed more potency (2-fold increase) than **9a** against *K. pneumonia* and showed the same activity as **9a** against *P. aeruginosa* and *S. typhimurium*, while showing a 2-fold decrease in activity than **9a** against *E. coli*. Introduction of the strongly electron withdrawing NO_2_ group as in compound **9f**, retained the broad spectrum of activity against Gram-positive and -negative organisms with improved potency (2-fold) against all strains except against *E. coli* (2-fold decrease in activity than **9a**). Replacement of phenyl group with benzyloxy group (**9g**) resulted in an improvement of antibacterial activity against all Gram-positive organisms, *S. epidermidis*, and *Micrococcus* spp. Although OBS **9g** showed the same antibacterial activity as compound **9a** against the Gram-negative organism *E. coli*, it exhibited moderate antibacterial activity against *K. pneumonia* and weak antibacterial activity against *P. aeruginosae* and *S. typhimurium*. Replacement of phenyl group with naphthyl group (**9h**) resulted in a loss of activity against all bacterial strains but surprisingly displayed the highest antifungal activity against *Aspergillus niger* and *Candida albicans* with MIC 8 and 4 µg/mL, respectively. Replacement of naphthyl group in **9h** with heterocyclic moieties led to retaining the activity against Gram-positive organisms only. On comparing these derivatives with compound **9a**, we noticed that introducing 2-furyl moiety (OBS **9i**) led to a 2-fold increase against the three test Gram-positive strains *S. aureus*, *S. epidermidis*, and *Micrococcus* spp. Installment of 2-thienyl group (**9j**) instead of the phenyl group displayed a decrease in the activity of nearly 4-fold against *Staphylococcus aureus* and 2-fold against *S. epidermidis*, *Micrococcus* spp. Shifting from phenyl group to 2-pyridyl group (**9k**) led to an increase in the antibacterial activity by 2-fold against *Micrococcus* spp. and retained the same activity as in compound **9a** against *S. aureus* and *S. epidermidis*.

##### Antifungal Activity of Synthesized Compounds

Notably, compound **9h** exhibited potent antifungal activity with MIC 4 and 2 µg/mL against *Aspergillus niger* and *Candida albicans*, respectively. Compound **9c** was second in potency compared with **9h,** and showed moderate antifungal activity with MIC 8 and 4 µg/mL against *Aspergillus niger* and *Candida albicans*, respectively. Compound **9k** showed moderate activity against *Candida albicans* and was weak against *Aspergillus niger* with MIC 8 and 16 µg/mL, respectively. Compound **9a** displayed weak antifungal activity against *Candida albicans* and very weak antifungal activity against *Aspergillus niger* with MIC 16 and 32 µg/mL, respectively. The rest of the compounds were inactive as antifungal agents, with MIC more than 32 µg/mL.

##### Antibiofilm Activity of Synthesized Compounds

Prior to the investigation of the anti-biofilm and anti-virulence activities of tested compounds, the effect of compounds at sub-MIC (=½ MIC) on *P. aeruginosa* and *S. aureus* growth was evaluated to exclude any effect on bacterial growth [49,50]. There were no significant differences between bacterial growth in the presence or absence of tested compounds at sub-MIC. It is worth mentioning that the sub-MIC concentrations of tested compounds are used in all the next experiments.

Bacterial virulence is regulated via a quorum sensing (QS) system in an inducer/receptor manner [51,52]. Diminishing bacterial virulence is an advantageous strategy to decrease the development of bacterial resistance [47,53,54,55,56]. In this direction, anti-virulence and anti-QS activities have been explored in several studies as reviewed [51]. The ability of *P. aeruginosa* or *S. aureus* to form biofilms was assayed in the absence or presence of tested compounds at sub-MIC. Significantly, most of the compounds were able to reduce the formation of biofilm, especially compounds **9a**, **9b** and **9f** (Figure 3A,B). The experiment was conducted in triplicate and a one-way ANOVA test was employed to test the statistical significance using Graphpad Prism 8 software. The results were significant statistically where *p* values < 0.05. Moreover, microscopic visualization of biofilm under the effect of tested compounds was also performed by light microscopy. Representative images for the inhibitory effects on *S. aureus* and *P. aeruginosa* biofilm formation are shown (Figure 3C,D). The microscopic images show a marked reduction in both the thickness of and surface area covered by the biofilms in presence of the tested compound.

##### Tested Compounds Diminish the Production of Bacterial Virulence Extracellular Enzymes

Bacteria establish their infection into the host cells by utilizing a diverse arsenal of virulence factors [57,58]. Extracellular enzymes play crucial roles in the bacterial invasion and spread, for instance, protease and hemolysins [48,53,54]. Herein, the proteolytic and hemolytic activities of selected compounds **9a**, **9b** and **9f** were assayed in *P. aeruginosa* and *S. aureus* (Figure 4). The tested compounds **9a**, **9b** and **9f** at sub-MIC showed significant ability to diminish the production of extracellular enzymes. The experiments were conducted in triplicate and the significance was analyzed using one-way ANOVA (Graphpad Prism 8 software). The results were significant statistically where *p* values <0.05. The results were presented as the percentage of inhibition from untreated bacterial.

##### Tested Compounds Diminish the Production of Bacterial Virulence Factors

*S. aureus* pigment staphyloxanthin is an important virulence factor due to antioxidant action that helps in overcoming the host immune defense [59]. Additionally, pyocyanin has emerged as an important virulence factor produced by *P. aeruginosa* [47,48]. The inhibitory effects of tested compounds **9a**, **9b** and **9f** at sub-MIC on the production of bacterial virulence factors staphyloxanthin in *S. aureus* and pyocyanin in *P. aeruginosa* were evaluated (Figure 5). The tested compounds showed a significant ability to reduce the production of bacterial pigments. The experiments were conducted in triplicate and the significance was analyzed using one-way ANOVA (Graphpad Prism 8 software). The results were significant statistically where *p* values < 0.05.

##### SAR Study

SAR of the synthesized candidates are summarized as follows (Figure 6):Introduction of 4-methoxy group (**9b**) resulted in a 2-fold increase in the activity against *Staphylococcus aureus*, *Staphylococcus epidermidis*, *Micrococcus* spp. and *Pseudomonas aeruginosae*, while retaining the same activity as compound **9a** against *Klebsiella pneumonia*, *Salmonella typhimurium* and *Escherichia coli*;Introduction of 2,5-dimethoxy group (**9c**) retained the same activity against *Staphylococcus epidermidis* and *Micrococcus* spp., and a 2-fold decrease against *Staphylococcus aureus*, *Pseudomonas aeruginosae*, *Klebsiella pneumonia*, *Salmonella typhimurium* and *Escherichia coli*;Replacement of hydrogen with trimethoxy groups (**9d**) resulted in the same activity against *Staphylococcus aureus*, improved activity (2-fold) against *Staphylococcus epidermidis* and *Micrococcus* spp., slightly decreased activity (2-fold) against *Escherichia coli*, and significantly decreased the antibacterial activity against *Pseudomonas aeruginosae*, *Klebsiella pneumonia*, and *Salmonella typhimurium*;Introduction of the weakly deactivating Cl group (**9e**) improved potency (2-fold increase in activity) against all Gram-positive organisms, *Staphylococcus epidermidis*, and *Micrococcus* spp., improved potency (2-fold increase) against *Klebsiella pneumonia*, resulted in the same activity against *Pseudomonas aeruginosae* and *Salmonella typhimurium*, and a 2-fold decrease in activity against *Escherichia coli;*Introduction of the strongly activating NO_2_ group (**9f**) improved potency (2-fold) against all strains except against *Escherichia coli* (2-fold decrease);Replacement of phenyl group with benzyloxy group (**9g**) led to the improvement of antibacterial activity against all Gram-positive organisms, the same antibacterial activity against the Gram-negative organism *Escherichia coli*, moderate antibacterial activity against *Klebsiella pneumonia*, and weak antibacterial activity against *Pseudomonas aeruginosae* and *Salmonella typhimurium*;Replacement of phenyl group with naphthyl one (**9h**) resulted in loss of activity against all bacterial strains, whereas it displayed the highest antifungal activity against *Aspergillus niger* and *Candida albicans*;Introduction of 2-furyl moiety (**9i**) led to a 2-fold increase against the three test Gram-positive strains *Staphylococcus aureus*, *Staphylococcus epidermidis* and *Micrococcus* spp.;Introduction of 2-thienyl group (**9j**) resulted in a decrease in activity by 4-fold against *Staphylococcus aureus*, and 2-fold against *Staphylococcus epidermidis* and *Micrococcus* spp.;Shifting from phenyl group to 2-pyridyl group (**9k**) increased the antibacterial activity by 2-fold against *Micrococcus* spp., while retaining the same activity against *Staphylococcus aureus* and *Staphylococcus epidermidis*.

Collectively, it could be concluded that the presence of unsubstituted phenyl group (**9a**), presence of one donating methoxy group (**9b**), or the presence of the strongly deactivation NO_2_ group (**9f**), is the optimum for antibacterial activity. On the other hand, it could be concluded that the presence of bulky naphthyl group (**9h**) is the optimum for antifungal activity. Replacement of naphthyl group with any other group resulted in either decreasing or abolishing the antifungal activity.

#### 2.2.2. The Antitumor Activity of the Tested Compounds

##### Cell Viability Assay

Firstly, to determine the cell viability, HPDE cell lines were treated with all new synthesized compounds **9a**–**k** for 96 h using sulforhodamine B (SRB) assay. All newly synthesized compounds were proven non-toxic with IC_50_ more than 50 mg/mL (Table 2).

##### Tested Compounds Suppress Cellular Proliferation of Cancer Cell Lines

The tested compounds’ effects on cellular proliferation of different cancer cell lines BxPC-3, Panc-1, HepG-2, and the normal immortalized cell line HPDE have been evaluated. SRB colorimetric assays have been conducted to evaluate cellular proliferation (Table 2). The tested compounds have varied anticancer activity ranging from moderate to very weak activity. Among all, compounds **9b** and **9f** displayed good anticancer activity against HepG2 cancer cell line with IC_50_ values = 8.53 and 6.39 µg/mL, respectively. Additionally, compound 9k exhibited good anticancer activity against PC3 cancer cell line with IC_50_ value = 7.27 µg/mL, in contrast with the normal HPDE cells that were the least affected after treatment (IC_50_ > 50 µg/mL). The remaining new synthesized compounds showed weak or very weak anticancer activity against the three cancer cell lines used.

##### Tested Compounds Can Induce Apoptotic Cell Death

Apoptotic cascade induction is a main chemotherapy-induced cell death procedure [60]. The apoptotic effects of selected compounds **9a**, **9b** and **9f** on pancreatic resistant cell lines and Panc-1 were assessed by quantification of caspase 3/7 levels (Figure 7). Our findings revealed that tested compounds triggered apoptosis through increasing the amounts of activated caspases 3/7 in Panc-1cell line compared with untreated controls. The apoptotic effect was dose-dependent, and the experiment was conducted in triplicate. The significance was analyzed using two-way ANOVA (Graphpad Prism 8 software) to compare the caspase levels at different concentrations. The results were significant statistically where *p* values < 0.05.

### 2.3. Docking Study into Pseudomonas aeruginosa QS Receptors

The process in which bacterial populations are controlled is called quorum sensing (QS) in which cells communicate with each other using signaling molecules called autoinducers that are produced by bacterial cells and detected by receptors on other bacterial cells. The QS signaling system orchestrates numerous physiological functions in both Gram-positive and Gram-negative bacteria [51,61]. Targeting bacterial virulence is a promising approach to decreasing the development of bacterial resistance [51,54]. In this approach, we used synthesized compounds at their sub-MIC which did not affect the bacterial growth and hence will not increase the possibility of resistance development [47,53]. In this context, it was necessary to evaluate the ability of tested compounds to antagonize the QS, which is the key regulator of bacterial virulence.

To explore the binding interactions and the capability of the most potent derivatives **9a**, **9b** and **9f** to antagonize the QS receptors, the interactions between **9a**, **9b**, **9f** and QS proteins were evaluated. *Escherichia coli* QS receptor was retrieved from the protein data bank (PDB: 1ROS) and molecular docking was carried out [62]. First, validation of the docking protocol was conducted by redocking of the ligand into the active site of the *Pseudomonas aeruginosa* QS receptor (PDB: 1ROS) (Figure 8). The RMSD value was less than 2 (0.835) which confirmed the validity of the docking results.

The results, as illustrated in Figure 8, displayed that derivatives **9a**, **9b** and **9f** were well-accommodated inside the binding cavity of the receptor. From the docking results, compound **9a** was incorporated into the formation of two hydrogen bonds, the oxygen of the sulfonamide group with Leu181, and the proton of sulfonamide nitrogen group with Pro238 amino acids. Additionally, it formed many hydrophobic interactions with Ile180, Leu181, His218, His222, His228 and Tyr240 amino acid residues (Figure 9A,B).

Similarly, compound **9b** engaged in the formation of two hydrogen bonds but with different amino acids, the oxygen of sulfonamide group with Ser230, and the carbonyl oxygen of oxazolone ring with Tyr240. **9b** formed many hydrophobic interactions with Ile180, Leu214, His218, Leu226 and His228 amino acid residues (Figure 9C,D).

Finally, compound **9f** was involved in the formation of three hydrogen bonds; the oxygen of sulfonamide group incorporated in the formation of two hydrogen bonds with Gly106, and Leu181, while the oxygen of NO_2_ group engaged in the third hydrogen bond with Ala182. Additionally, compound **9f** was involved in many hydrophobic interactions with His172, Ile180, His183, His218, Thr239 and Tyr240 amino acid residues (Figure 9E,F). These results are almost in agreement with the biological evaluation and may explain the possible reasons for enhanced anti-QS activity of compounds **9a**, **9b** and **9f,** suggesting these three compounds for further study as novel promising antibiofilm and antimicrobial candidates.

## 3. Materials and Methods

### 3.1. Chemistry

Melting points were determined with a Gallenkamp (London, UK) melting point apparatus and are uncorrected. IR spectra (KBr, cm^−1^) were recorded on Bruker Vector, 22FT-IR (Fourier Transform Infrared (FTIR), Ettlingen, Germany) spectrometer. Unless otherwise specified, proton (^1^H) and carbon (^13^C) NMR spectra were recorded at room temperature in base filtered (CD_3_)_2_SO on a spectrometer operating at 400 MHz for proton and 100 MHz and 300 MHz for proton and 75 MHz for carbon nuclei. The signal due to residual (CH3)2SO appearing at δ H 2.5 and the central resonance of the (CD_3_)_2_SO “multipet” appearing at δ C 39.0 were used to reference ^1^H and ^13^C NMR spectra, respectively. 1H NMR data are recorded as follows: chemical shift (δ) (multiplicity, coupling constant(s) J (Hz), relative integral) where multiplicity is defined as s = singlet, d = doublet, t = triplet, q = quartet, and m = multiplet or combinations of the above. Elemental analyses were determined using a manual elemental analyzer Heraeus (Germany) and an automatic elemental analyzer CHN Model 2400 Perkin Elmer (USA) at Microanalytical Center, Faculty of Science, Cairo University, Egypt. All the results of elemental analyses corresponded to the calculated values within experimental error. Progress of the reaction was monitored by thin-layer chromatography (TLC) using precoated TLC sheets with ultraviolet (UV) fluorescent silica gel (Merck 60F254, Merck, Darmstadt, Germany), and spots were visualized by iodine vapors or irradiation with UV light (254 nm). All chemicals were purchased from Sigma-Aldrich or Lancaster Synthesis Corporation (Welwyn Garden, UK). Intermediate **3** [63] was prepared according to reported procedure.

#### 3.1.1. General Procedure for the Synthesis of 2-((4-Methylphenyl)sulfonamido)benzoic acid **3**

Anthranilic acid **1** (0.10 mmol) was dissolved in 30 mL sodium hydroxide (2 N) in a 500 mL conical flask. The mixture was stirred vigorously with a mechanical stirrer until the solid was almost completely dissolved. 4-toluenesulfonyl chloride **2** (0.10 mmol) was added in five portions and stirred vigorously for a further 1 h. The crystallized 4-toluenesulfonyl anthranilic was left in the refrigerator overnight. The crystals were filtered on a Buchner funnel, washed with ice cold water and dried at 100 °C. The product was crystallized from ethanol [63].

#### 3.1.2. General Procedure for the Synthesis of *N*-(2-(1H-Benzo[d][1,2,3]triazole -1-carbonyl) phenyl)-4-methylbenzenesulfonamide **5**

Thionyl chloride (0.08 mL, 1 mmol) was added to a solution of 1*H*-benzotriazole **4** (0.48 g, 0.4 mmol) in DCM (10 mL) at room temperature, the reaction mixture was stirred for 20 min, then acid **3** (0.2 g, 1 mmol) was added to the reaction mixture, which was stirred for 3 h at 25 °C. The reaction was diluted with DCM (50 mL) then the organic layer was washed with saturated Na_2_CO_3_ (3 × 20 mL), H_2_O (2 × 20 mL) and brine (1 × 10 mL), then dried (sodium sulfate), and filtered. Hexane (50 mL) was added to the filtrate, the obtained solid was dried under reduced pressure to give compound 5, which was crystallized from ethanol.

Yellowish solid, yield (89%); m.p. 190–192 °C. ^1^H NMR (400 MHz, DMSO-d_6_) δ: 2.27 (s, 3H), 7.04 (d, *J* = 8.4 Hz, 1H), 7.24 (d, *J* = 8.0 Hz, 2H), 7.38 (t, *J* = 7.6 Hz, 1H), 7.45 (d, *J* = 8.0 Hz, 2H), 7.56–7.50 (m, 1H), 7.66 (t, *J* = 7.2 Hz, 1H), 7.78 (dd, *J* = 7.6, 1.6 Hz, 1H), 7.85 (t, *J* = 8.2 Hz, 1H), 8.29 (t, *J* = 8.2 Hz, 2H), 10.06 (s, 1H) ppm. ^13^C NMR (100 MHz, DMSO) δ: 20.9, 114.3, 119.9, 125.2, 125.5, 126.4, 126.6, 128.6, 129.4, 130.5, 131.3, 131.3, 132.62, 135.28, 136.15, 143.22, 145.35, 165.72 ppm. Anal. Calcd for C_20_H_16_N_4_O_3_S: C, 61.21; H, 4.11; N, 14.28. Found: C, 61.39; H, 4.17; N, 14.45.

#### 3.1.3. General Procedure for the Synthesis of (2-((4-Methylphenyl)sulfonamido) benzoyl)glycine **7**

To a solution of benzenesulfonamide derivative **5** (0.31 g, 1 mmol) in acetonitrile (5 mL), a solution of glycine **6** (0.11 g, 1.5 mmol) in acetonitrile/H_2_O (7/3 mL) and triethylamine (0.12 mL, 1 mmol) was added. The reaction mixture was stirred at 25 °C for 12 h then monitored by TLC. After completion of the reaction, 6 N HCl (1 mL) was added, the reaction mixture was concentrated under reduced pressure. The residue thus obtained was partitioned between H_2_O (20 mL) and ethyl acetate (20 mL), and the separated organic layer was washed with 4 N HCl (3 × 5 mL) and brine (10 mL), then dried (MgSO_4_), filtered and concentrated under reduced pressure to deliver acid **6**. The product was crystallized from ethanol.

Yellow solid, yield (80%); m.p. 198–200 °C. ^1^H NMR (300 MHz, DMSO-d_6_) δ: 2.31 (s, 3H), 3.91 (d, *J* = 6 Hz, 2H), 7.10 (t, *J* = 7.2 Hz, 1H), 7.34 (t, *J* = 8.4 Hz, 2H), 7.43–7.57 (m, 2H), 7.63–7.74 (m, 2H), 7.89 (d, *J* = 7.5 Hz, 1H), 9.13 (s, 1H), 11.48 (s, 1H), 12.74 (s, 1H) ppm. ^13^C NMR (75 MHz, DMSO-d6) δ: 20.9, 41.2, 116.5, 118.3, 119.1, 126.8, 128.4, 129.8, 132.7, 135.8, 138.5, 139.9, 143.7, 144.0, 168.6, 170.7 ppm. Anal. Calcd for C_16_H_16_N_2_O_5_S: C, 55.16; H, 4.63; N, 8.04. Found; C, 54.89; H, 4.67; N, 7.93.

#### 3.1.4. General Procedure for the Synthesis of **9a**–**j**

A mixture of (2-((4-methylphenyl)sulfonamido)benzoyl)glycine **7** (0.30 g, 1.10 mmol) and the appropriate aldehydes **8a**–**j** (1.00 mmol) in acetic anhydride (1 mL) and fused sodium acetate (0.1 g, 1.2 mmol) was heated in an oil bath at 80 °C for 2 h. After cooling down at room temperature the mixture was allowed to stand for 24 h at 0 °C. The precipitate was filtered off and washed three times with ice-cooled ethanol (10 mL). The product was crystallized from ethanol.

#### 3.1.5. (E)-*N*-(2-(4-Benzylidene-5-oxo-4,5-dihydrooxazol-2-yl)phenyl)-4-methylbenzenesulfonamide **9a**

Whitish solid, yield (87%); m.p. 214–216 °C. ^1^H NMR (300 MHz, DMSO-d_6_) δ 2.32 (s, 3H, CH_3_), 7.21–7.25 (m, 2H, ArH), 7.35 (d, *J* = Hz, 2H, ArH), 7.48 (s, 1H, ArH), 7.48–7.46 (m, 6H, ArH), 7.78 (d, *J* = Hz, 2H, ArH), 7.88 (d, *J* = Hz, 1H, CH=C), 11.57 (s, 1H, NH) ppm. ^13^C NMR (75 MHz, DMSO-d_6_) δ 20.9, 111.1, 117.5, 123.5, 126.9, 128.4, 129.1, 129.9, 130.0, 130.9, 131.1, 131.7, 132.0, 132.9, 134.8, 135.6, 139.0, 144.3, 162.8, 164.9 ppm. Anal. Calcd for C_23_H_18_N_2_O_4_S: C, 66.02; H, 4.34; N, 6.69. Found; C, 65.99; H, 4.28; N, 6.92.

#### 3.1.6. (E)-*N*-(2-(4-(4-Methoxybenzylidene)-5-oxo-4,5-dihydrooxazol-2-yl)phenyl)-4-methylbenzenesulfonamide **9b**

Straw yellow solid, yield (88%); m.p. 190–192 °C.1H NMR (300 MHz, DMSO-d_6_) δ: 2.32 (s, 3H, CH3), 3.89 (s, 3H, OCH_3_), 7.09 (d, *J* = 9 Hz, 2H, ArH), 7.18–7.28 (m, 1H, ArH), 7.36 (d, *J* = 8.1 Hz, 2H, ArH), 7.46 (s, 1H, ArH), 7.61 (d, *J* = 7.6 Hz, 2H, ArH), 7.77 (d, *J* = 8.1 Hz, 2H, ArH), 7.86 (d, *J* = 7.8 Hz, 1H, CH=C), 8.23 (d, *J* = 9.0 Hz, 2H, ArH), 11.58 (s, 1H, NH) ppm. ^13^C NMR (75 MHz, DMSO-d6) δ: 20.9, 56.0, 106.8, 117.5, 120.8, 121.3, 121.9, 123.3, 123.6, 126.7, 126.9, 127.1, 129.7, 130.0, 130.1, 130.3, 130.6, 135.3, 139.0, 144.4, 150.4, 161.8, 173.1 ppm. Anal. Calcd for C_24_H_20_N_2_O_5_S: C, 64.27; H, 4.50; N, 6.25. Found: C, 64.53; H, 4.53; N, 6.34.

#### 3.1.7. (E)-*N*-(2-(4-(2,5-diMethoxybenzylidene)-5-oxo-4,5-dihydrooxazol-2-yl)phenyl)-4-methylbenzenesulfonamide **9c**

White solid, yield (85%); m.p. 220–222 °C. ^1^H NMR (300 MHz, DMSO-d_6_) δ: 2.32 (s, 3H, CH_3_), 3.85 (s, 3H, OCH_3_), 3.91 (s, 3H, OCH_3_), 7.13–7.25 (m, 2H, ArH), 7.26–7.34 (m, 3H, ArH), 7.61 (s, 2H, ArH), 7.72 (d, *J* = 7.8 Hz, 2H, ArH), 7.85 (d, *J* = 7.8 Hz, 2H, ArH), 7.95 (d, *J* = 6.1 Hz, 1H, CH=C), 10.92 (s, 1H, NH) ppm. ^13^C NMR (75 MHz, DMSO-d_6_) δ: 20.9, 55.7, 56.4, 112.6, 113.1, 114.9, 118.9, 120.4, 120.5, 121.5, 124.0, 124.6, 126.9, 129.9, 130.0, 130.6, 134.5, 135.6, 138.3, 153.3, 153.6, 162.6, 165.3 ppm. Anal. Calcd for C_25_H_22_N_2_O_6_S: C, 62.75; H, 4.63; N, 5.85. Found; C, 62.89; H, 4.67; N, 5.93.

#### 3.1.8. (E)-4-Methyl-*N*-(2-(5-oxo-4-(3,4,5-trimethoxybenzylidene)-4,5-dihydrooxazol-2-yl)phenyl)be-nzenesulfonamide **9d**

Yellowish solid, yield (78%); m.p. 230–232 °C. ^1^H NMR (300 MHz, DMSO-d_6_) δ: 2.31 (s, 3H, CH_3_), 3.80 (s, 3H, OCH_3_), 3.92 (s, 6H, 2OCH_3_), 7.15–7.35 (m, 3H, ArH), 7.41 (s, 1H, ArH), 7.49–7.68 (m, 6H, ArH), 7.86 (d, *J* = 7.8 Hz, 1H, CH=C), 10.84 (s, 1H, NH) ppm. ^13^C NMR (75 MHz, DMSO-d_6_) δ: 20.9, 55.9, 56.2, 60.3, 107.5, 109.9, 111.9, 116.4, 119.6, 124.3, 126.9, 128.2, 129.9, 132.1, 134.4, 135.3, 138.1, 144.2, 153.1, 162.8, 165.9 ppm. Anal. Calcd for C_26_H_24_N_2_O_7_S: C, 61.41; H, 4.76; N, 5.51. Found; C, 61.44; H, 5.02; N, 5.62.

#### 3.1.9. (E)-*N*-(2-(4-(4-Chlorobenzylidene)-5-oxo-4,5-dihydrooxazol-2-yl)phenyl)-4-methylbenzenesulfonamide **9e**

Yellow solid, yield (85%); m.p. 234–236 °C. ^1^H NMR (300 MHz, DMSO-d_6_) δ 2.33 (s, 3H, CH_3_), 7.24–7.38 (m, 3H, ArH), 7.57–7.69 (m, 3H, ArH), 7.79–7.94 (m, 3H, 2ArH + CH=C), 8.32 (d, *J* = 8.6 Hz, 2H, ArH), 8.45 (d, *J* = 7.8 Hz, 2H, ArH), 11.48 (s, 1H, NH) ppm. ^13^C NMR (75 MHz, DMSO-d_6_) δ 20.9, 111.1, 117.6, 123.5, 127.0, 129.1, 129.4, 129.9, 130.1, 131.4, 131.8, 133.4, 134.9, 135.6, 136.2, 139.0, 144.3, 163.1 ppm. Anal. Calcd for C_23_H_17_ClN_2_O_4_S: C, 61.00; H, 3.78; N, 6.19. Found; C, 60.78; H, 3.67; N, 6.39.

#### 3.1.10. (E)-4-Methyl-*N*-(2-(4-(4-nitrobenzylidene)-5-oxo-4,5-dihydrooxazol-2-yl)phenyl)benzenesulfonamide **9f**

Pale yellow solid, yield (88%); m.p. 250–252 °C. ^1^H NMR (300 MHz, DMSO-d_6_) δ: 2.31 (s, 3H, CH_3_), 7.26 (t, *J* = 7.8 Hz, 1H, ArH), 7.37 (d, *J* = 8.1 Hz, 2H, ArH), 7.53–7.72 (m, 3H, ArH), 7.80 (d, *J* = 7.8 Hz, 2H, ArH), 7.92 (d, *J* = 8.1 Hz, 1H, CH=C), 8.32 (d, *J* = 8.4 Hz, 2H, ArH), 8.45 (d, *J* = 8.7 Hz, 2H, ArH), 11.48 (s, 1H, NH) ppm. ^13^C NMR (75 MHz, DMSO-d_6_) δ: 20.9, 111.0, 117.7, 119.0, 123.8, 126.9, 127.2, 129.2, 129.8, 130.3, 130.6, 132.5, 133.2, 135.4, 135.5, 139.2, 140.2, 144.4, 148.0, 164.5, 165.5 ppm. Anal. Calcd for C_23_H_17_N_3_O_6_S: C, 59.61; H, 3.70; N, 9.07. Found; C, 59.83; H, 3.79; N, 9.18.

#### 3.1.11. (E)-*N*-(2-(4-(4-(Benzyloxy)benzylidene)-5-oxo-4,5-dihydrooxazol-2-yl)phenyl)-4-methylbenzenesulfonamide **9g**

Whitish solid, yield (88%); m.p. 235–237 °C. 1H NMR (300 MHz, DMSO-d_6_) δ: 2.32 (s, 3H, CH_3_), 5.25 (s, 2H, OCH_2_-) 7.18–7.37 (m, 3H, ArH), 7.35–7.50 (m, 8H, ArH), 7.61 (d, *J* = 6.8 Hz, 2H, ArH), 7.77 (d, *J* = 8.3 Hz, 2H, ArH), 7.87 (d, *J* = 7.8 Hz, 1H, CH=C), 8.23 (d, *J* = 8.8 Hz, 2H, ArH), 11.60 (s, 1H, NH) ppm. ^13^C NMR (75 MHz, DMSO) δ: 168.62, 162.64, 161.75, 161.36, 160.13, 144.38, 138.69, 136.45, 135.58, 134.56, 134.37, 131.51, 130.12, 129.74, 128.50, 128.05, 127.85, 126.99, 125.94, 123.61, 115.60, 111.40, 69.60, 20.97 ppm. Anal. Calcd for C_30_H_24_N_2_O_5_S: C, 68.69; H, 4.61; N, 5.34. Found: C, 68.66; H, 4.64; N, 5.38.

#### 3.1.12. (E)-4-Methyl-*N*-(2-(4-(naphthalen-1-ylmethylene)-5-oxo-4,5-dihydrooxazol-2-yl)phenyl)benzenesulfonamide **9h**

White solid, yield (76%); m.p. 226–228 °C. ^1^H NMR (300 MHz, DMSO-d_6_) δ: 2.31 (s, 3H, CH_3_), 7.25-7.36 (m, 3H, ArH), 7.62-7.75 (m, 7H, ArH), 7.76-8.42 (m, 3H, ArH), 8.45 (d, *J* = 7.8 Hz, 2H, ArH), 8.75 (d, *J* = 7.5 Hz, 1H, CH=C), 11.60 (s, 1H, NH) ppm. ^13^C NMR (75 MHz, DMSO-d_6_) δ: 21.00, 118.87, 119.5, 123.3, 127.0, 127.2, 128.8, 129.1, 129.9, 131.5, 132.3, 132.4, 133.3, 134.0, 135.9, 138.8, 143.9, 165.9, 168.3 ppm. Anal. Calcd for C_27_H_20_N_2_O_4_S: C, 69.22; H, 4.30; N, 5.98. Found; C, 68.98; H, 4.07; N, 5.92.

#### 3.1.13. (E)-*N*-(2-(4-(Furan-2-ylmethylene)-5-oxo-4,5-dihydrooxazol-2-yl)phenyl)-4-methylbenzenesulfonamide **9i**

Yellow solid, yield (84%); m.p. 191–193 °C. ^1^H NMR (300 MHz, DMSO-d_6_) δ: 2.33 (s, 3H, CH_3_), 7.08–7.12 (m, 2H, ArH), 7.12-7.32 (m, 3H, ArH), 7.36–7.47 (m, 3H, ArH), 7.51–7.88 (m, 3H, ArH), 7.90 (d, *J* = 8.4 Hz, 1H, CH=C) ppm, exchangeable 1H due to NH. ^13^C NMR (75 MHz, DMSO-d_6_) δ: 20.9, 116.8, 118.2, 118.8, 122.9, 123.1, 126.7, 126.8, 128.9, 129.8, 129.9, 131.5, 132.8, 134.3, 135.8, 139.2, 140.0, 143.7, 144.0, 169.7 ppm. Anal. Calcd for C_21_H_16_N_2_O_5_S: C, 61.76; H, 3.95; N, 6.86. Found; C, 61.89; H, 3.67; N, 6.93.

#### 3.1.14. (E)-4-Methyl-*N*-(2-(5-oxo-4-(thiophen-2-ylmethylene)-4,5-dihydrooxazol-2-yl)phenyl)benzenesulfonamide **9j**

Yellow solid, yield (90%); m.p. 196–198 °C. ^1^H NMR (300 MHz, DMSO-d_6_) δ: 2.32 (s, 3H, CH3), 7.24–7.35 (m, 1H, ArH), 7.38–7.50 (m, 2H, ArH), 7.58–7.63 (m, 1H, ArH), 7.77–7.87 (m, 4H, ArH), 7.90–8.22 (m, 3H, ArH), 8.24 (d, *J* = 7.6 Hz, 1H, CH=C), 11.49 (s, 1H, NH) ppm. ^13^C NMR (75 MHz, DMSO-d_6_) δ: 20.9, 111.1, 117.7, 123.5, 125.3, 126.9, 128.0, 128.7, 129.8, 130.0, 134.6, 135.6, 136.2, 136.8, 137.2, 138.6, 144.3, 161.3, 164.3 ppm. Anal. Calcd for C_21_H_16_N_2_O_4_S_2_: C, 59.42; H, 3.80; N, 6.60. Found; C, 59.61; H, 3.87; N, 6.82.

#### 3.1.15. (E)-4-Methyl-*N*-(2-(5-oxo-4-(pyridin-4-ylmethylene)-4,5-dihydrooxazol-2-yl)phenyl)benzenesulfonamide **9k**

Whitish solid, yield (78%); m.p. 210–212 °C. ^1^H NMR (300 MHz, DMSO-d_6_) δ: 2.32 (s, 3H, CH_3_), 7.21–7.38 (m, 5H, ArH), 7.59–8.19 (m, 7H, ArH), 8.20 (d, *J* = 7.8 Hz, 1H, CH=C), 11.27 (s, 1H, NH) ppm. ^13^C NMR (75 MHz, DMSO-d_6_) δ: 20.9, 111. 7, 117.7, 123.5, 125.3, 126.9, 128.7, 129.8, 130.0, 134.5, 135.3, 135.8, 136.2, 137.0, 138.7, 144.0, 163.3, 165.7 ppm. Anal. Calcd for C_22_H_17_N_3_O_4_S; Calcd C, 63.00; H, 4.09; N, 10.02. Found: C, 63.31; H, 4.16; N, 10.21.

### 3.2. Biological Activity

#### 3.2.1. Evaluation of Antimicrobial and Anti-Virulence Activities

##### Determination of Minimum Inhibitory Concentration (MIC)

The MICs of the synthesized compound were determined by agar dilution method according to the Clinical Laboratory and Standards Institute Guidelines (CLSI, 2015) [55,64]. Briefly, the tested strains were incubated overnight in tryptic soy broth (TSB) (Oxoid, United Kingdom) and then diluted in Muller–Hinton (MH) broth (Oxoid, United Kingdom) to turbidity approximating to the equivalent of 0.5 McFarland standard [65]. The suspensions were further diluted with sterile saline (1:10) and standardized inoculums (approximately 10^4^ CFU per spot) were spotted on the surfaces of MH agar (Oxoid, United Kingdom) plates containing different concentrations of tested compounds and a control plate. The MICs were the lowest concentrations that inhibit growth on the plates after incubation at 37 °C for 20 h.

##### Excluding the Effect of Compounds on Bacterial Growth

To avoid any expected effect of tested compound on the bacterial virulence, the effect of compounds at their sub-MIC (½ MIC) on bacterial growth was evaluated [50,58]. The tested strains Pseudomonas aeruginosa ATCC 47,085 and Staphylococcus aureus ATCC 6538 were grown in Luria–Bertani (LB) Broth (Oxoid, Hampshire, United Kingdom) overnight at 37 °C in presence of tested compounds at sub-MIC (½ MIC). The experiment was conducted in triplicate, and the optical densities of bacterial growth were compared with control untreated bacteria. It is worth mentioning that the tested compounds were used at sub-MIC (½ MIC) in all the next performed tests to evaluate the anti-virulence activities.

##### Assay of Biofilm Formation

In order to evaluate the ability of the tested compounds to inhibit the biofilm formation, a strong biofilm forming *P. aeruginosa* ATCC 47,085 [47,66] and *S. aureus* ATCC 6538 [67] strains were used. As described earlier [68,69], suspensions of tested strains were prepared from overnight cultures in TSB and their optical densities were adjusted to OD600 of 0.4 (1×10^8^ CFU/mL). Aliquots of 10 μL of the suspensions were added to 1 mL amounts of fresh TSB with or without sub-MICs of tested compounds. Then, 100 μL of TSB with or without tested compounds in sub-MIC were transferred into the wells of 96-well microtiter plates and incubated at 37 °C overnight. The non-adherent cells were removed, the wells were washed with sterile PBS, and left to dry. The attached biofilm forming cells were fixed with methanol for 25 min and stained with 1% crystal violet for 30 min. The excess dye was washed out and the crystal violet staining adhered biofilm forming cells were eluted by glacial acetic acid (33%). The experiment was conducted in triplicate and the absorbance was measured at 590 nm. The absorbances of tested strains treated with different compounds were expressed as mean ± standard error of percentage change from untreated tested strains control. The percentages of biofilm inhibition were calculated employing the following formula: (absorbance of control—absorbance in presence of tested compounds)/absorbance of control.

##### Assay of Protease Production

The effect of compounds **5**, **7** and **10** on the production of protease was evaluated using casein substrate as described earlier [47,48]. Briefly, overnight cultures of *P. aeruginosa* and *S. aureus* were cultivated in LB broth in the presence or absence of compounds **5**, **7** and **10** at ½ MIC for 24 h at 37 °C. The supernatants were collected, mixed (1:1) with 0.05 M casein in phosphate buffer (2%) and NaOH (0.1 M) at pH 7.0, and incubated for 15 min at 37 °C. The reaction was stopped by adding 2 mL of 0.4 M trichloroacetic acid for 30 min at 25 °C. Any precipitates were removed, and the optical densities were detected at 660 nm. The assays were performed in triplicate and the obtained optical densities of tested strains treated with compounds **5**, **7** and **10** were expressed as mean ± standard error of percentage change from untreated tested strains control (positive control) and LB (negative control). The protease inhibition percentages were calculated: (O.D control—O.D tested compounds)/O.D control.

##### Assay of Hemolytic Activity

The anti-virulence effects of compounds **5**, **7** and **10** on hemolytic activity of tested *P. aeruginosa* and *S. aureus* strains were assessed as described previously [52,53]. Optically adjusted bacterial cultures treated or untreated with tested compounds at sub-MIC were centrifuged, and 0.5 mL of supernatants were mixed with fresh 0.8 mL 2% erythrocyte (obtained from experimental animals) suspension in saline, and incubated for 2 h at 37 °C. A complete hemolysis positive control was prepared by addition of sodium dodecyl sulphate (SDS) to erythrocyte suspension, and negative control was prepared by incubation of erythrocytes in LB broth under the same conditions. After centrifugation, the absorbances of the lysed erythrocytes were measured at 540 nm by Biotek spectrofluorometer (Biotek, Winooski, VT, USA). The experiment was performed in triplicate, and the hemolysis of tested compound treated cultures were expressed as mean ± standard error of percentages compared with those obtained from untreated control cultures using the formula: (absorbance in presence or absence of tested compounds—absorbance of negative control)/(absorbance of positive control—absorbance of negative control).

##### Quantification of Staphyloxanthin Pigment

Staphyloxanthin and intermediate carotenoids were extracted from *S. aureus* treated or untreated with compounds **5**, **7** and **10** at sub-MIC as described [59]. Bacterial cells were cultivated in TSB at 37 °C for 24 h, then cells were collected by centrifugation, and washed twice with phosphate-buffered saline (PBS). The obtained pellets were used to extract staphyloxanthin with methanol. The pellets (5 gm of) were resuspended in 20 mL methanol, and heated with gentle stirring at 55 °C in a water bath for 5 min. Then, the methanol extract liquids were cooled and centrifuged, and the absorbances of the produced staphyloxanthin were quantified spectrophotometrically at 450 nm (Biotek, Winooski, VT, USA). The experiment was repeated in triplicate, and the pigment absorbances in the presence of tested compounds were expressed as mean ± standard error of percentage change from untreated controls. The percentages of pigment production were calculated using the formula: (absorbance of control − absorbance in presence of tested compounds)/absorbance of control.

##### Quantification of Pyocyanin Pigment

The ability of selected compounds **5**, **7** and **10** at sub-MIC to reduce the *P. aeruginosa* pyocyanin pigment production was estimated as described earlier [47,48]. *P. aeruginosa* overnight cultures were prepared and diluted in LB broth at 600 nm (O.D0.4), and 10 μL of the bacterial suspensions were added to 1mL broth tubes containing, or not, tested drugs at sub-MIC. After incubation for 48 h at 37 °C, the tubes were centrifuged and the pyocyanin in the supernatant was assayed spectrophotometrically at 691 nm by a Biotek spectrofluorometer (Biotek, Winooski, VT, USA). The experiment was repeated in triplicate, and the pyocyanin absorbances in the presence of tested compounds were expressed as mean ± standard error of percentage change from untreated controls. The percentages of pigment production were calculated using the formula: (absorbance of control − absorbance in presence of tested compounds)/absorbance of control.

#### 3.2.2. Evaluation of Antitumor Activities of Synthesized Compounds

##### Effect of Synthesized Compounds on Cellular Proliferation

The pancreatic human cancer cell lines BxPC-3 and Panc-1, the human hepatocellular carcinoma (HepG-2), and the normal immortalized pancreatic cell line HPDE, that were used in this study, were obtained from the American Type Culture Collection (Rockville, USA). Cell lines were cultured and treated with the tested compounds or dimethyl sulfoxide (DMSO) as previously described [70,71]. Cells were cultivated in DMEM medium (Invitrogen, Carlsbad, CA, USA), supplemented with streptomycin, penicillin and fetal bovine serum (FCS) (Invitrogen, Carlsbad, CA, USA).

The sulforhodamine B (SRB) assay was employed to assess the anti-proliferative effects of tested compounds on cancer cells [60,72]. Cell lines were incubated and regularly treated with DMSO or increasing doses of tested compounds for 48 h. The cells were fixed with 10% trichloroacetic acid and stained with SRB fluorescent dye for 30 min. Then, the bounded SRB dye to cellular proteins was dissolved in 10 mM Tris base after washing excess dye with 1% acetic acid. The absorbance was measured at 510 nm in a reader (Biotek, Winooski, VT, USA).

##### Evaluation of Caspase-3/7 Activity

Caspases play critical roles in apoptosis. The apoptotic effects of selected compounds 5, 7 and 10 on pancreatic cell line Panc-1 were tested by quantification of caspase 3/7 using Caspase-Glo 3/7 assay kit (Promega, Fitchburg, MA, USA) as previously described [60,71]. Briefly, cells were treated with or without compounds **5**, **7** and **10** (½ IC50, IC50, or 2 ×IC50) for 6 h. The prepared reagent Caspase-Glo 3/7 was added in equal volumes to cells, gently mixed, and incubated for 60 min at room temperature. The luminescence was measured, and the activity of caspase was presented as a percentage change from the untreated control.

### 3.3. In Silico Docking Study

Docking simulation study was carried out using Discovery Studio 2.5 software (Accelrys Inc., San Diego, CA, USA) [73]. For more details, see Appendix A.

## 4. Conclusions

In summary, eleven oxazolone-benzenesulfonamide compounds **9a–k,** were synthesized and characterized by IR, NMR (^1^H and ^13^C) and elemental analyses. The title compounds were evaluated for their in vitro antibiofilm, antimicrobial and anticancer activities. The majority of the tested compounds displayed potent antibacterial activity against both Gram-positive and -negative bacteria. Compounds **9a**, **9b** and **9f** exhibited considerable antibacterial activity. Compound **9h** exhibited the most potent antifungal activity. Compounds **9a**, **9b**, **9f** and **9k** showed good anticancer activity against different cancer cell lines. Importantly, several synthesized compounds showed a significant ability to inhibit the formation of biofilm by *Pseudomonas aeruginosa* and *Staph. aureus*. The compounds **9a**, **9b** and **9f** displayed the most potent antibiofilm inhibition activity; that is why these three compounds were subjected to further investigation for their anti-virulence activities. The three compounds **9a**, **9b** and **9f** significantly reduced the production of QS-controlled virulence factors. These findings are in great compliance with the ability to hinder the QS receptors in silico, indicating that these compounds can serve as anti-QS and anti-virulence agents.

## Data Availability

Not applicable.

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
