# Peer review of "Synthesis, Antimicrobial, Anti-Virulence and Anticancer Evaluation of New 5(4H)-Oxazolone-Based Sulfonamides"

_molecules, 2022, doi:10.3390/molecules27030671_

Round 1

Reviewer 1 Report

This manuscript reports on the synthesis of eleven oxazolone-benzenesulfonamide compounds and they were evaluated in various in vitro and docking studies

Within these studies, specifically those of antibacterial activity, it is emphasized that compounds 9a, 9b, and 9f were the ones that exhibited the best performance. On page 7, lines 167-171, it is described that both 9a and 9b, which even with the introduction of the electron-donating group (4-MeO, 9b) retain the same activity as 9a against Klebsiella pneumonia, Salmonella typhimurium, and Escherichia coli. In the case of 9f (the introduction of the strong electron-withdrawing group NO2) it presented a broad spectrum of activity against all strains except E. coli.

Is there any explanation why in the case of 9b and 9f, even when they are so different electronically speaking due to their functional groups, they present such outstanding antimicrobial activities when compared to the other compounds? This is important to discuss since for this reason so many compounds were synthesized to contrast with each other.

Why in the case of antifungal studies did compound 9h show the best activity over the rest? Did the presence of the naphthyl group have any influence on the rest of the compounds? Correct typing and other errors: 

1) Page 7, line 167: the beginning of the sentence is incomplete

2) In point 2.2.1.6, in numbers (1, 2, 3, 4, 6, 7 y 10) correct the use of &.

3) Page 15, line 394: ligan

4) Page 13, line 339: weal

 In this way, the authors are recommended to correct and explain these observations for the article to be accepted.

Author Response

Dear Reviewer 1,

We are very grateful for your valuable comments and suggestion. Please find the attached reply to all raised points. 

Best Regards,

Reviewer 2 Report

The manuscript (Manuscript ID: molecules-1559916) “Synthesis, antimicrobial, anti-virulence and anticancer evaluation of new 5(4H)-Oxazolone-Based Sulfonamides” submitted by Ahmad J. Almalk et al. presents attempt of the studies on a new 5(4H)-oxazolone-benzene sulfonamide derivatives. The proposed subject can raise some interest in scientific community. The manuscript is well written and is generally clear. However, it requires some revision to remove typos and some editorial errors. After a little improvement (minor revisuin), the article qualifies for publication.

The following critical remarks should be made on this work:

  • The figure 3 is no needed. All all chemical formulas of tested compounds are clearly presented for Scheme 2.
  • Which values are in Table 2? Are the data correspond to mean values (±SD) for n≥3?
  • In the docking results, it should be noted which functional groups interact with the ligand.
  • Figures 9-10 are also not clear, especially in Fig. 10 capital letters should be larger. Separate colors used for different interactions should be added to the figure caption.

Author Response

Dear Reviewer 2,

We are very grateful for your valuable comments and suggestion. Please find the attached reply to all raised points. 

Best Regards,

Reviewer 3 Report

This manuscript describes synthesis and biological activities of 5(4H)-oxazolone-based sulfonamide compounds.  I think the results of the biological activities are interesting.  Therefore, this manuscript would be suitable to publish as an article in Molecules.  However, there are tons of writing mistakes.  The authors should check the manuscript carefully before resubmitting.

Change the style of H of 5(4H) to italic throughout the manuscript.

Compound number should be written in bold style throughout the manuscript.

Is the reaction condition from 5 and 6 to 7 right?

Scientific names of microorganisms should be written in italic style throughout the manuscript.

Which of Aspragillus niger and Aspergillus niger is correct?

Hyphen should be inserted between locant and prefix in all cases.

Author Response

Dear Reviewer 3,

We are very grateful for your valuable comments and suggestion. Please find the attached reply to all raised points. 

Best Regards,
